# Effects on Quality of Life and Self-Efficacy of Instant Messaging Services in Self-Management Programs for Prostate Cancer: A Systematic Review and Meta-Analysis

**DOI:** 10.3390/cancers17030465

**Published:** 2025-01-29

**Authors:** Julia Raya Benítez, Geraldine Valenza Peña, Javier Martín Núñez, Alba Navas Otero, María Granados Santiago, Alejandro Heredia Ciuró, Marie Carmen Valenza

**Affiliations:** 1Department of Nursing, University of Granada, 18071 Granada, Spain; juliarb@ugr.es (J.R.B.); mariagranados@ugr.es (M.G.S.); 2Department of Physiotherapy, University of Granada, 18071 Granada, Spain; geraldinevalenza@ugr.es (G.V.P.); albanavas@ugr.es (A.N.O.); ahc@ugr.es (A.H.C.); cvalenza@ugr.es (M.C.V.)

**Keywords:** self-management, instant messaging, quality of life, self-efficacy, systematic review, meta-analysis

## Abstract

Instant messaging platforms offer a growing opportunity for effective therapeutic support, but evidence on their role in self-management remains limited. The aim was to observe the effects of self-management interventions based on instant messaging on quality of life and self-efficacy in patients diagnosed with prostate cancer. The results seem to show positive effects of self-management programs based on instant messaging on self-efficacy, as well as improved quality of life, in prostate cancer patients.

## 1. Introduction

Prostate cancer stands among the most prevalent cancers among men worldwide [1]. The EUROCARE-5 study reported a 5-year relative survival rate for prostate cancer in Europe of 83.4% for the 2000–2007 period [2]. Remarkably, significant improvements in 5-year relative survival rates for prostate cancer have been observed [3]. Although a significant portion of the increase in prostate cancer survival in the general population can be attributed to the rise in diagnoses of tumors with favorable prognoses resulting from the widespread use of the prostate-specific antigen (PSA) tests, flaws in population studies may have influenced the observed data among various countries. Survival in prostate cancer has been additionally demonstrated to be associated with age [4], comorbidities [5], and treatment [6].

Multiple management options are available for men diagnosed with prostate cancer. These include active surveillance for men with less aggressive prostate cancer, surgery and radiation for localized disease, and chemotherapy for metastatic disease [7]. When considering the long-term treatments of prostate cancer survivors, various behavior programs have demonstrated significant results by emphasizing individuals’ perceptions, beliefs, and awareness [8]. Through these programs, which are frequently in the form of self-management, individuals are more likely to engage in health promotion behaviors, perceive the severity of risks, view proposed behaviors as beneficial, and overcome barriers to adoption [9,10].

Mobile health interventions are used for disseminating health information and improving health outcomes [11]. E-health interventions include PC-based interventions, internet-based interventions [12], and mobile phone-based interventions, including instant messaging services. The user base for instant messaging is expanding rapidly, with the growing number of active daily users allowing for the implementation of easy-to-use therapeutic interventions.

SMS-based therapeutic education has proven effective in promoting screening behaviors in various studies [13,14]. However, research on the efficacy of self-management interventions, particularly those utilizing instant messaging tools, remains limited. Therefore, this study aims to evaluate the effects of self-management interventions based on instant messaging on quality of life and self-efficacy.

## 2. Methods and Materials

The Preferred Reporting Items for Systematic Reviews and Meta-Analyses (PRISMA) statement guidelines were followed to conduct this systematic review and meta-analysis. (Appendix A) The Cochrane Collaboration guidelines were used as a reference, and we also registered our protocol in the PROSPERO database with the code CRD42021292127.

### 2.1. Search Strategy

The search strategy used was developed for MEDLINE, taking into account: (1) keywords and key terms used in other existing systematic reviews on the same topic; (2) the use of MeSH terms after a thorough examination of the database; and (3) the assistance of a specialist in guiding and reviewing the database. Our search equation was adapted to the PubMed, Web of Sciences, and Scopus databases to allow for a broad search of articles indexed up to November 2024. Finally, the reference lists of previous reviews were taken into account, and the search strategy was refined in the various databases mentioned by means of search tests.

### 2.2. Inclusion and Exclusion Criteria

Based on the participants, interventions, outcomes, and study design, a research question was formulated based on the PICOS model [15]. Based on this model, the inclusion criteria used were: (1) patients with a diagnosis of prostate cancer; (2) self-management interventions based on instant messaging; (3) self-management interventions based on instant messaging compared with a control or no-treatment intervention group; (4) self-efficacy and quality of life as outcome study variables; and (5) randomized clinical trials.

Our review is based on the Practical Reviews in Support of Self-Management (PRISMS) definition for the selection of self-management trials. This taxonomy includes 14 components (e.g., information about condition/management, provision of equipment, social support…) that support self-management interventions in a variety of settings. This taxonomy allows for the classification of self-management strategies in people with chronic conditions or their caregivers, taking into account the general dimensions of mode of delivery, personnel providing support to the person, focused intention, intensity, and frequency and duration of the intervention [16]. The selection of instant messaging interventions was made while taking into account the previously published taxonomy of collaborative technologies. With respect to the control group intervention, other interventions that do not include self-management based on instant messaging interventions will be considered [17].

### 2.3. Literature Data Extraction

After the process of obtaining records from the different databases was completed, the process of eliminating duplicate studies was carried out. Then, two reviewers independently performed the selection of studies based on the titles and abstracts of the studies to ensure their eligibility. In case of discrepancies, a third reviewer resolved the inclusion doubts.

### 2.4. Literature Quality Assessment

After the selection of studies, data extraction and methodological assessment of the studies were performed again independently by two reviewers and a third reviewer who resolved discrepancies [18]. Methodological quality was assessed using the Downs and Black checklist. This scale consists of five subscales into which the 27 items are divided (“reporting”, “external validity”, “internal validity (study bias, confounding)”, “selection bias”, and “study power”). This scale evaluates the fulfillment of each item, which makes it possible to classify the methodological quality as excellent if the score is above 26, good if the score is between 20 and 25, fair if the score is between 1 and 19, and poor if the score is below 14. The Downs and Black scale has high validity and reliability, making it a widely used tool in randomized clinical trials [19,20].

### 2.5. Risk of Bias Analysis

The Cochrane risk of bias tool allowed us to assess the risk of bias for each of the clinical trials included in our review. Following its guidelines, we assessed the risk of bias of the subscale’s selection bias performance, detection bias, attrition bias, reporting bias, and other bias. Based on the risk of bias of each of these subscales, we obtained a final score that allowed us to classify the included studies into low risk of bias (if all subscales were at low risk of bias), some concern of bias (if one or two subscales were unclear), or high risk of bias (if one scale had limitations that invalidated the results or if more than two subscales were unclear) [21].

### 2.6. Meta-Analysis

Review Manager 5 (RevMan software 5.4.1) was used to conduct a quantitative synthesis for studies reporting mean values and standard deviations related to quality of life and self-efficacy. To calculate the overall mean differences (MDs) between the experimental and control groups, data on the number of participants assessed, final mean values, and standard deviations for each group were extracted.

When studies lacked sufficient data to calculate effect sizes, such as missing mean values or standard deviations, the authors were contacted for clarification. Standard mean differences (SMDs) were used for all scales because, although they measured the same underlying symptom or condition, the outcome measures were different. Depending on the statistical heterogeneity indicated by I^2^ tests, either random-effects models or fixed-effects models were applied; fixed-effects models were chosen when heterogeneity was below 50%. Forest plots were created to visualize the overall impacts of the interventions [21]. Forest plots were visually inspected for outlier studies, source heterogeneity was explored, and sensitivity analyses were conducted by excluding trials that had the most influence on the results.

### 2.7. Qualitative Analyses

The GRADE system (Grading of Recommendations, Assessment, Development, and Evaluation) was utilized to qualitatively analyze and classify the evidence levels of the studies reviewed. This process involved evaluating five critical domains: study design, precision, relevance, consistency, and the potential for publication bias [22].

Each of the domains was assessed using the GRADE criteria, as follows: (1) study design—scores were penalized if there was uncertainty or significant risk of bias, or if there were limitations in effect estimation; (2) inconsistency—scores were penalized if point estimates varied significantly between studies or if there was high heterogeneity; (3) indirectness—scores were penalized if there were significant differences between interventions, study populations, or outcomes; and (4) imprecision—scores were penalized if the sample size was insufficient (*n* ≥ 400) [23,24]. In addition, the effect size calculated in the meta-analysis was taken into account to increase the validity of the assessment of the certainty of the evidence.

Qualitative analysis of the manuscript was performed by two authors with extensive experience in cancer (JMV and AHC). This analysis was performed independently by the reviewers, and a third author (MCV) resolved disagreements between them to reach consensus on the GRADE scale.

## 3. Results

Figure 1 presents the flowchart of the search, screening, and selection stages of the study process. The initial search identified a total of 19,984 records. After removing duplicates, 13,442 records remained. Screening based on the title and abstract resulted in the selection of 462 articles. Out of these, 24 reports could not be retrieved, so 438 reports were assessed for eligibility. Following full-text evaluation, eight articles [25,26,27,28,29,30,31,32] were included in the qualitative synthesis and six studies in the quantitative synthesis [26,27,28,29,30,31].

The characteristics of the participants are detailed in Table 1. A total of 687 prostate cancer patients were included in this review, with age ranges from 60 to 85 years in the control group and 60 to 80 years in the experimental group. Most studies did not report the etiology of the cancer. Concerning cancer stage, three studies [27,28,31] included patients from T1 to T4, and one study included patient in the advanced stage (T2–T4) [31] and others in early stages (T1–T3) [32]. Only one study included patients in the T4 stage [30]. Two studies [25,26] did not report the stage of the participants.

The patients’ treatment statuses presented in the included studies were heterogenous. The study of Lawen et al. [25] delivered the intervention before cancer treatment, three studies [26,27,30] included patients during cancer treatment, and one study [32] applied the intervention after oncology treatment. The study of Evans et al. [29] conducted the interventions at various treatment times. Cancer treatments included surgery, radiotherapy, chemotherapy, and hormone therapy. Four studies [26,27,28,30] applied isolated hormone therapy, and one study [25] used surgery, radiotherapy, and/or hormone therapy. Evans et al. [29] also included patients who were receiving chemotherapy.

Upon applying the Cochrane risk of bias assessment, six studies had some concerns of bias and two studies had low risks of bias (Figure 2). With regard to the quality of the studies, the scores ranged from 21 to 22 according to the Downs and Black checklist, with the exception of the study by Langlais et al. [28], which presented poor quality, with a score of 14 points.

The characteristic interventions of the included studies are presented in the Table 2. Self-management interventions were classified using the PRIMS taxonomy. The component most commonly used in the studies was “education about condition and/or its management”, which was applied in all intervention groups of the studies [25,26,27,28,29,30,31,32]. Other applied components were “training for practical self-management activities” [25,26,29] “regular clinical review” [27,29,30], “provision of equipment” [28,32], and “information about available resources” [29,30]. Some components were less used: “provision of specific actions plans and rescue medication” [26,30], “social support” [25], training and rehearsal for psychological strategies” [27], “practical support with adherence” [28], and “monitoring condition with feedback” [30].

The instant messaging received by the intervention groups consisted of text messages (TM), short messaging service (SMS), and instant messaging applications (IMapp). Implementation was heterogeneous. Combinations of the different modalities were applied by different studies. Three studies combined IMapp with SMS [26,27,29], and one study [25] blended IMapp with TM. The remained studies used only one modality of instant messaging, with TM [28,31,32] being the most common reported. Only the study by Yang et al. [30] used IMapp as the modality.

The control group received usual care [26,27,29,30,32] or self-management [27,28,31]. The self-management component that participants in these studies received was “education about condition and/or its management”.

The length of the intervention ranged from 1.5 to 6 months. Most of the studies lasted three months. However, two studies had shorter intervention periods, particularly 6 weeks [27] and two months [29]. The study of Lawen et al. [25] performed a longer intervention, reaching 6 months.

Quality of life and self-efficacy were the outcomes assessed in this review. In relation to quality of life, the tools used for assessing it were heterogeneous. The main questionnaires applied were the EPIC and EORTC QLQ-C30. After applying the self-management intervention to instant messaging, only three studies [25,26,31] showed significant improvements in quality of life. The study of Park et al. [31] showed statistically significant differences in favor of the experimental group regarding global quality of life (*p* < 0.001). Two studies [25,26] reported significant differences (*p* < 0.001) in favor of the experimental group in the urinary irritation/obstruction subscale. The results for quality of life are also shown in Table 2. Regarding self-efficacy outcomes, we also found that the tools used to evaluate this variable were heterogeneous; however, all the studies showed significant improvements after the intervention when the experimental group was compared with the control. This was also the case with respect to the baseline measurement, except for the study by Yang et al. [30], which did not show a significant intragroup improvement when comparing the measurements before and after treatment.

### 3.1. Results of the Meta-Analyses

Figure 3 presents the results of the meta-analysis of the quality of life of prostate cancer patients, comparing the results regarding self-management achieved through instant messaging against the control group.

The pooled mean difference (MD) did not show significant overall effects when self-management achieved through instant messaging interventions was compared to the control group (MD = 0.16; 95% CI = −0.03, 0.36; *p* = 0.10). The results did not show heterogeneity (I^2^ = 0%).

Figure 4 presents the results of the meta-analysis regarding the self-efficacy of prostate cancer patients, comparing the results regarding self-management achieved through instant messaging against the control group.

The pooled mean difference (MD) did not show significant overall effects when self-management achieved through instant messaging interventions was compared to the control group (MD = 0.80; 95% CI = −0.03, 1.63; *p* = 0.06). The results did not show high heterogeneity (I^2^ = 95%).

### 3.2. Results of the GRADE

We applied the GRADE recommendations (Figure 5) to evaluate the level of evidence for the use of self-management achieved through instant messaging in prostate cancer and obtained a high recommendation for quality of life and a low recommendation for self-efficacy. The decrease in the assessment of certainty in self-efficacy was primarily due to inconsistency, as the meta-analyses showed moderate-to-high heterogeneity (95%), and was secondarily due to imprecision (n = 230).

## 4. Discussion

The objective of this systematic review was to evaluate the efficacy of self-management interventions based on instant messaging on quality of life and self-efficacy. Our results showed that self-management interventions based on instant messaging had positive effects on quality of life and self-efficacy when compared to a control group that did not use the instant messaging strategy.

Our results are in line with previous reviews such as that of Zou et al. [33], who observed significant effects on quality of life and self-efficacy when instant messaging was applied to cancer patients. In addition, other previous studies have already specifically observed positive results of educational and self-management programs using the Internet or smart devices in prostate cancer patients [34,35,36].

In this systematic review, all the included studies carried out interventions based on self-management, but included instant messaging among their strategies. These messages allow patients to motivate themselves for participation and improve their autonomy [37]. Therefore, providing adequate information and encouragement from healthcare professionals can help to maintain concrete behavior and motivate patients [38].

This was shown in the results obtained, with statistically significant improvements in self-efficacy observed in the included studies when self-management interventions based on instant messaging were followed; however, in our meta-analysis, although improvements were found in favor of self-management interventions based on instant messaging, they were not significant. Previous systematic reviews have already found positive effects of self-management programs on prostate cancer patients’ self-efficacy [39,40], but including instant messaging as one of the strategies seems to have a positive effect on these programs. Instant messaging supports self-management programs by maximizing participants’ motivation and providing information and guidance, and is, therefore, a good option for improving self-efficacy in prostate cancer patients [41,42].

In terms of quality of life, not all the trials included in our systematic review showed significant results after the self-management interventions based on instant messaging [29,32]. Of the trials that assessed quality of life specifically for prostate cancer-related symptoms, only Lee et al. [26] showed significant improvements in all aspects. In the meta-analysis of quality of life, although there were positive effects in favor of the experimental group, these were not significant. In line with the results of previous reviews [43,44], it was found that there is a great deal of heterogeneity in the relationship between symptoms and quality of life in prostate cancer patients after treatment, and that it is, therefore, necessary to assess this over time after the interventions. We also agree with Langlais et al. [28] that the sample included individuals at different stages of disease, and the studies varied in the time since diagnosis and/or treatment, which could have influenced the results obtained.

When interpreting the results, it is important to consider the sample size of each included study and the duration of the interventions. Previous reviews regarding self-management have highlighted the need to take these factors into account when interpreting the results [44,45,46]. In our case, this is not entirely clear, because studies such as Evans et al. [29], where the sample size was small (n = 38) and the duration of the intervention was 2 months, did not find statistically significant improvements in quality of life or in self-efficacy in the experimental group compared to baseline. However, other studies with even smaller sample sizes, such as Park et al. (n = 21) [31], found significant results for quality of life. This may be due not only to the sample size and duration of the intervention, but also to the frequency with which the included sample received feedback and instant messaging. In this case, the study by Evan et al. [29] only presented a two-month intervention with once-monthly feedback, which may have influenced the results.

Therefore, new lines of research should focus their studies on observing the optimal time and frequency of intervention. It would also be interesting for future reviews to compare the results between the different forms of intervention in order to resolve the heterogeneity that was found.

### Limitations

This systematic review has several limitations that need to be pointed out in order to interpret our results. Firstly, our results are positive in favor of self-management interventions based on instant messaging; however, no statistically significant improvements were found in the meta-analyses performed for quality of life and self-efficacy, and the heterogeneity present in the analysis of self-efficacy makes it necessary to interpret our results with caution. As for heterogeneity, it should be taken into account that it was also present in the tools used to assess the quality of life and self-efficacy variables, which may possibly have influenced our results. Finally, although all the studies utilized instant messaging as a self-management strategy in their interventions, the rest of the self-management strategies used in their designs were not the same. This could mean that, although instant messaging seems to be a good strategy and has positive effects on quality of life and self-efficacy, no significant improvements were found in our meta-analysis; therefore, future studies could study isolated effects of the instant messaging strategy in the prostate cancer population.

Our review also presents certain strengths that should be highlighted, as the results of our study make a novel contribution by focusing on the use of instant messaging as a complement to self-management strategies, highlighting its potential to improve patient participation and autonomy. Although the heterogeneity of the studies means that conclusions should be drawn with caution, this also allows us to improve the generalizability of the findings regarding this messaging strategy. In addition, the use of established frameworks in taxonomies such as PRIMS and GRADE recommendations strengthens the reliability of our findings.

## 5. Conclusions

In conclusion, this systematic review highlights the potential benefits of self-management interventions incorporating instant messaging for improving quality of life and self-efficacy in prostate cancer patients. Although the results of our meta-analyses do not demonstrate statistically significant overall effects, the observed positive trends underscore the value of these interventions in supporting patient autonomy and quality of life.

## Figures and Tables

**Figure 1 cancers-17-00465-f001:**
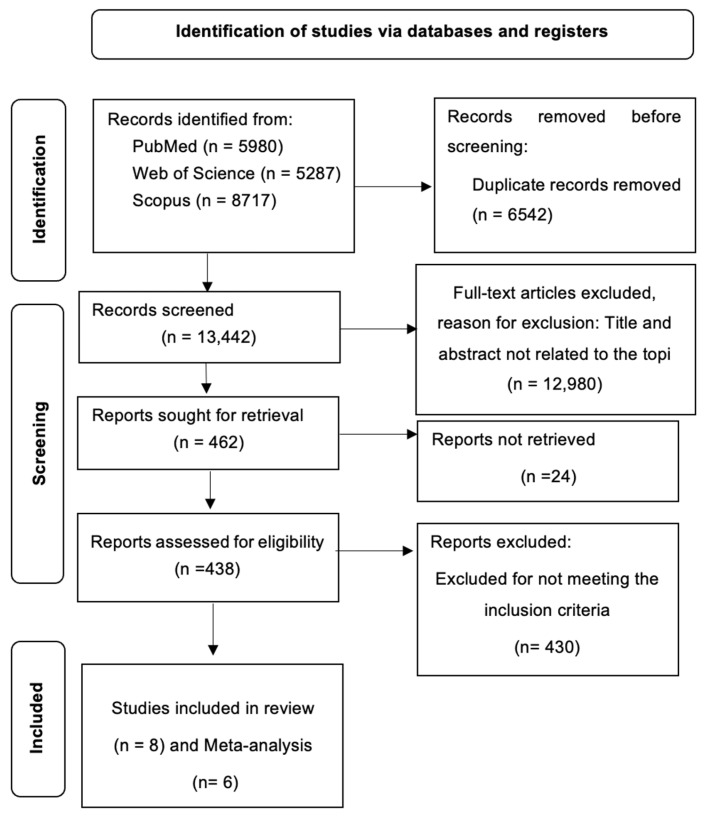
Flowchart of the selected studies.

**Figure 2 cancers-17-00465-f002:**
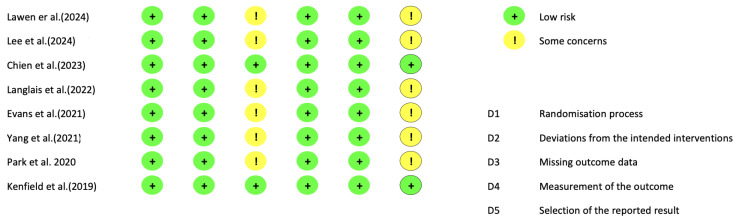
Risk of bias assessment of the included studies [25,26,27,28,29,30,31,32].

**Figure 3 cancers-17-00465-f003:**
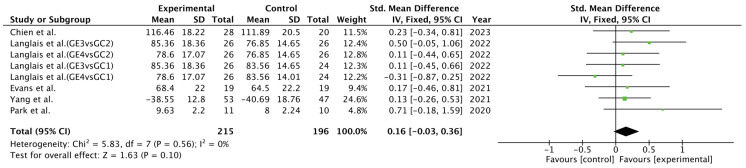
Forest plot for quality of life [27,28,29,30,31].

**Figure 4 cancers-17-00465-f004:**
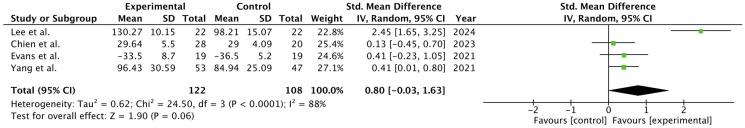
Forest plot for self-efficacy [26,27,29,30].

**Figure 5 cancers-17-00465-f005:**
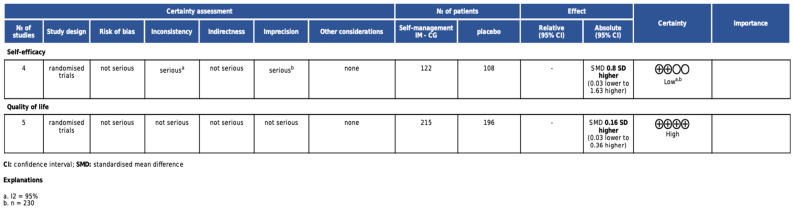
GRADE results.

**Table 1 cancers-17-00465-t001:** Characteristics of the included studies.

Study	TNM Cancer Stage	Treatment Status, Timing	Sample Size (IG/CG)	Sample Age	Quality Assessment Downs & Black
Lawen et al. (2024) [25]	NR	Pre-treatment (surgery, radiotherapy, and/or hormone therapy)	66/62	IG: 66 (60, 70) CG: 68 (61, 72)	22 (Good Quality)
Lee et al. (2024) [26]	NR	During treatment (hormone therapy)	22/24	IG: 67.59 ± 6.59 CG: 69.96 ± 7.48	21 (Good Quality)
Chien et al. (2023) [27]	T1–T4	During treatment (hormone therapy)	36/40	IG: 73.22 ± 7.54 CG: 76.83 ± 8.33	22 (Good Quality)
Langlais et al. (2022) [28]	T1–T4	Post-treatment (hormone therapy)	50 + 52/49 + 51	70 (65, 75)	14 (Poor Quality)
Evans et al. (2021) [29]	T4	Pre-, during, and post-treatment (surgery, radiotherapy, chemotherapy, and hormone therapy)	19/19	IG: 69.5 ± 6.6 CG: 70.8 ± 10.2	21 (Good Quality)
Yang et al. (2021) [30]	T2–T4	During treatment (hormone therapy)	53/47	IG: 66.11 ± 7.92 CG: 67.94 ± 5.87	21 (Good Quality)
Park et al. (2020) [31]	T1–T4	NR	11/10	IG: 66 (61, 71) CG: 67 (59.5, 73)	22 (Good Quality)
Kenfield et al. (2019) [32]	T1–T3	Post-treatment (NR)	37/39	IG: 66 (61, 68) CG: 65 (60, 69)	21 (Good Quality)

**Table 2 cancers-17-00465-t002:** Characteristics of interventions.

Study	Intervention Group	Instant Messaging	Comparation Group	Trial Length	Quality of Life
Lawen et al. (2024) [25]	1. Education about condition and/or its management 2. Training for practical self-management activities 3. Social support 4. Instant messaging	TM: daily and weekly multimedia content, daily practice reminders IMapp: monthly conferences and to-do lists	1. Education about condition and/or its management	6 months	EPIC-26 Urinary incontinence: Pre < Post (*p* < 0.001) EG < CG (*p* < 0.001) Urinary irritation/obstruction Pre < Post (*p* < 0.001) EG < CG (*p* < 0.001) Sexual health (*p* = 0.4): Pre < Post (NS) EG < CG (NS) Bowel: Pre < Post (NS) EG < CG (NS) Hormonal: Pre < Post (NS) EG < CG (NS)
Lee et al. (2024) [26]	1. Education about condition and/or its management 2. Provision on specific action plans and rescue medication 3. Training for practical self-management activities 4. Instant messaging	IMapp: weekly information and multimedia content SMS: feedback	Usual care	3 months	EPIC-26 Urinary incontinence: Pre < Post (*p* < 0.05) EG < CG (*p* < 0.05) Urinary irritation/obstruction: Pre < Post (*p* < 0.001) EG < CG (*p* < 0.001) Sexual health: Pre < Post (*p* < 0.05) EG < CG (*p* < 0.05) Bowel: Pre < Post (*p* < 0.05) EG < CG (*p* < 0.05) Hormonal: Pre < Post (*p* < 0.05) EG < CG (*p* < 0.05) LET: Pre < Post (*p* < 0.001) EG > CG (*p* < 0.001)
Chien et al. (2023) [27]	1. Education about condition and/or its management 2. Regular clinical review 3. Training and rehearsal for psychological strategies 4. Instant messaging	IMapp: information, weekly feedback SMS: weekly reminders	Usual care	6 weeks	FACT-G: Pre < Post (*p* < 0.05) EG < CG (*p* < 0.05) FACT-P: Pre < Post (*p* < 0.05) EG < CG (*p* < 0.05) GSES: Pre < Post (*p* < 0.05) EG < CG (*p* < 0.05)
Langlais et al. (2022) [28]	IG1: 1.Education about condition management 2. Provision of equipment 3. Web resources 4. Instant messaging IG2: 1.Education about condition management 2. Provision of equipment 3. Web resources 4. Practical support with adherence 5. Instant messaging	TM: information four times a week	CG1: 1. Education about condition management 2. Web resources CG2: 1. Education about condition management 2. Web resources	3 months	EORTC QLQ-C30: Pre < Post (*p* < 0.05) EG < CG (*p* < 0.05)
Evans et al. (2021) [29]	1. Education about condition and/or its management 2. Information about available resources 3. Regular clinical review 4. Training for practical self-management activities 5. Instant messaging	IMapp: two conferences, feedback SMS: assistance four times	Usual care	2 months	EORTC QLQ-C30: Pre < Post (NS) EG < CG (NS) BS: Pre > Post (NR) EG < CG (*p* < 0.07)
Yang et al. (2021) [30]	1. Education about condition and/or its management 2. Information about available resources 3. Provision of specific clinical action plans 4. Regular clinical reviews 5. Monitoring of condition with feedback 6. Instant messaging	IMapp: information and multimedia content twice a month, feedback: messages, conferences	Usual care	3 months	AMS: Pre > Post (*p* < 0.05) EG > CG (*p* < 0.05) SUPPH: Pre < Post (NS) EG > CG (*p* < 0.001)
Park et al. (2020) [31]	1. Education about condition and its management 2. Instant messaging	TM: motivational information three times a week	1. Education about condition and/or its management	3 months	EORTC QLQ-C30: Pre < Post (*p* < 0.01) EG < CG (*p* < 0.01)
Kenfield et al. (2019) [32]	1. Education about condition and/or its management 2. Provision of equipment 3. Instant messaging	TM: information twice a month, support and feedback several times a week	Usual care	3 months	EPIC-26 Urinary incontinence: Pre < Post (NS) EG < CG (NS) Urinary irritation/obstruction: Pre < Post (NS) EG < CG (NS) Sexual health: Pre < Post (NS) EG < CG (NS) Bowel: Pre < Post (NS) EG < CG (NS) Hormonal: Pre < Post (NS) EG < CG (NS) SF-36: Pre < Post (NS) EG < CG (NS)

Mean ± SD; median [IQR], mean change (95% IC). AMS: Anging Male’s Symptoms Scale; EORTC QLQ-30: European Organization for Research and Treatment of Cancer (EORTC) Quality of Life Questionnaire-30 (QLQ-C30); EPIC-26: Expandex Prostate Cancer Index Composite-26; FACT-P: Functional Assessment of Cancer Therapy-Prostate; GSES: The General Self-Efficacy Scale. Chinese version; LET: Lifestyle Evaluation Tool; IMapp: Instant Messages app; SF-36: Short Form Health Survey-36; SUPPH: strategies used by people to promote health; TM: text messages.

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
