# Peer review of "Effects on Quality of Life and Self-Efficacy of Instant Messaging Services in Self-Management Programs for Prostate Cancer: A Systematic Review and Meta-Analysis"

_cancers, 2025, doi:10.3390/cancers17030465_

Round 1

Reviewer 1 Report

Comments and Suggestions for Authors

Very Respected Authors,

After reading your paper I have few suggestions. Paper is well-organized. The objective of the paper is clear. The methodology is described in detail and the results are well presented. In addition to the limitations, any advantages, if present, should also be mentioned. In addition to the limitations, any advantages, if present, should also be mentioned. It is necessary to expand the conclusion in both the abstract and the paper.

Author Response

Please, find enclosed a revision of our manuscript, “Effects on Quality of Life and Self-efficacy of Instant Messaging Services in Self-Management Programs for Prostate Cancer: A Systematic Review and Meta-Analyses.”. We would like to thank the Editorial committee for giving us the opportunity to revise and improve our manuscript, and the Editors and reviewers for their thoughtful and constructive comments. We have considered all of the suggestions and have incorporated them into the revised manuscript. All changes to the manuscript have been highlighted or indicated by using tracked changes, and we believe our manuscript is stronger as a result of these modifications. An itemized point-by-point response to the reviewers’ comments is presented below.

Reviewer 1

Very Respected Authors,

Comment 1: After reading your paper I have few suggestions. Paper is well-organized. The objective of the paper is clear. The methodology is described in detail and the results are well presented. In addition to the limitations, any advantages, if present, should also be mentioned. It is necessary to expand the conclusion in both the abstract and the paper.

Response: We appreciate your comments and congratulations. In relation to your corrections we have included a section on advantages and we have expanded our conclusion in order to improve the quality of our study.

Reviewer 2 Report

Comments and Suggestions for Authors

Dear Authors,

The manuscript requires several improvements to meet the publication standards. First, I recommend addressing the high similarity index reported by iThenticate, which is currently at 49%. This level of similarity is not acceptable for original research and must be reduced to align with publication requirements.

The application of the GRADE method to the Summary of Findings (SoF) table is unclear in the current manuscript. This is a critical methodological aspect because it directly affects the reliability and interpretability of the presented evidence. Specifically, the manuscript does not provide information about the qualifications or background of the individuals who performed the GRADE assessment. Were the ratings conducted by experts in systematic reviews, biostatistics, or the clinical field relevant to prostate cancer? Understanding the expertise of the assessors is crucial for evaluating the credibility of the evidence appraisal. Additionally, it is important to clarify whether the ratings were conducted independently by multiple reviewers and, if so, how disagreements were resolved. If consensus meetings or third-party adjudication were employed, this process should also be detailed. These elements should be explicitly described in the Methods section to enhance transparency and to allow readers to assess the robustness of the grading process.

Finally, there is a significant concern about the meta-analysis. The use of mean differences is problematic since the outcome measures in the included studies were assessed using different scales. To address this, standardized mean differences should be employed instead, as they are better suited to account for varying measurement tools across studies. This change is essential to ensure the validity and robustness of the meta-analytic findings.

Minor point: Regarding the PRISMA flow diagram, if the section labeled “Identification via other methods” contains all zeros, it can be removed to simplify and clarify the diagram. This would improve the visual presentation of the study selection process.

I hope these suggestions help improve the quality of the manuscript.

Author Response

Please, find enclosed a revision of our manuscript, “Effects on Quality of Life and Self-efficacy of Instant Messaging Services in Self-Management Programs for Prostate Cancer: A Systematic Review and Meta-Analyses.”. We would like to thank the Editorial committee for giving us the opportunity to revise and improve our manuscript, and the Editors and reviewers for their thoughtful and constructive comments. We have considered all of the suggestions and have incorporated them into the revised manuscript. All changes to the manuscript have been highlighted or indicated by using tracked changes, and we believe our manuscript is stronger as a result of these modifications. An itemized point-by-point response to the reviewers’ comments is presented below.

Reviewer 2

Dear Authors,

Comment 1: The manuscript requires several improvements to meet the publication standards. First, I recommend addressing the high similarity index reported by iThenticate, which is currently at 49%. This level of similarity is not acceptable for original research and must be reduced to align with publication requirements.

Response: Thank you for your appreciation and we have taken the opportunity to rewrite the methodology section in order to reduce this percentage. Sometimes we use our own articles as a basis for the methodology of new manuscripts since they follow similar structures. However, we have modified the entire section

Comment 2: The application of the GRADE method to the Summary of Findings (SoF) table is unclear in the current manuscript. This is a critical methodological aspect because it directly affects the reliability and interpretability of the presented evidence. Specifically, the manuscript does not provide information about the qualifications or background of the individuals who performed the GRADE assessment. Were the ratings conducted by experts in systematic reviews, biostatistics, or the clinical field relevant to prostate cancer? Understanding the expertise of the assessors is crucial for evaluating the credibility of the evidence appraisal. Additionally, it is important to clarify whether the ratings were conducted independently by multiple reviewers and, if so, how disagreements were resolved. If consensus meetings or third-party adjudication were employed, this process should also be detailed. These elements should be explicitly described in the Methods section to enhance transparency and to allow readers to assess the robustness of the grading process.

Response: We thank you for your comment and we have taken the opportunity to clarify what you have indicated in the methodology and in the authors' contribution section. The qualitative analysis was carried out by authors JMN and AHC who have contributed different articles related to cancer to the scientific community, and specifically systematic reviews in which the methodological process of the GRADE scale was carried out for the qualitative analysis of the included studies. This ensures the methodological quality on the part of these authors, in addition a third author MCV participated to solve the discrepancies and supervise its correct realization (this author presents a great amount of articles in relation to cancer and systematic reviews in other fields of chronic pathologies).

Comment 3: Finally, there is a significant concern about the meta-analysis. The use of mean differences is problematic since the outcome measures in the included studies were assessed using different scales. To address this, standardized mean differences should be employed instead, as they are better suited to account for varying measurement tools across studies. This change is essential to ensure the validity and robustness of the meta-analytic findings.

Response: We appreciate your comment and have therefore made changes in the methodology and results section with respect to our meta-analysis. First, we have adapted our methodology to the analysis performed (mean difference). In relation to the results, the meta-analysis of the quality of life variable shows significant results when comparing only the mean difference, however, when observing the weight of each of the articles we observed that Park et al. had a considerable influence, so a sensitivity analysis was carried out to ensure that our results are interpreted correctly and do not present bias.

Comment 4: Minor point: Regarding the PRISMA flow diagram, if the section labeled “Identification via other methods” contains all zeros, it can be removed to simplify and clarify the diagram. This would improve the visual presentation of the study selection process.

Response: We appreciate your comment and agree with what you say. For this reason we have modified figure 1 and we have eliminated the part that you indicate to make it easier for the readers to understand and to make the presentation of our article cleaner.

Reviewer 3 Report

Comments and Suggestions for Authors

Thank you for the review of your well written manuscript. There is a clear background leading into your method.  The method is clear and provides details of the review process and analysis.

The results and tables are well supported with details and provide an easy to understand structure. 

The discussion summarizes the findings.  It would be good to include aspects related to the sample size and differences in length of times of the intervention.  I think this adds to the discussion about the small sample sizes and differences in intervention length.  Was there a difference in the significance with the longer studies or no change? 

Adding research and clinical recommendations would be a useful addition.  

Conclusion is only one sentence. Maybe add details of what may make a good study based on your findings.  

Author Response

Please, find enclosed a revision of our manuscript, “Effects on Quality of Life and Self-efficacy of Instant Messaging Services in Self-Management Programs for Prostate Cancer: A Systematic Review and Meta-Analyses.”. We would like to thank the Editorial committee for giving us the opportunity to revise and improve our manuscript, and the Editors and reviewers for their thoughtful and constructive comments. We have considered all of the suggestions and have incorporated them into the revised manuscript. All changes to the manuscript have been highlighted or indicated by using tracked changes, and we believe our manuscript is stronger as a result of these modifications. An itemized point-by-point response to the reviewers’ comments is presented below.

Reviewer 3

Comment 1: Thank you for the review of your well written manuscript. There is a clear background leading into your method.  The method is clear and provides details of the review process and analysis. The results and tables are well supported with details and provide an easy to understand structure.

Response: We appreciate your comments and thanks and have taken into account the rest of your comments to improve our manuscript.

Comment 2: The discussion summarizes the findings.  It would be good to include aspects related to the sample size and differences in length of times of the intervention.  I think this adds to the discussion about the small sample sizes and differences in intervention length.  Was there a difference in the significance with the longer studies or no change?

Response: Thank you for your comment and take this opportunity to include it and modify our discussion section.

Comment 3: Adding research and clinical recommendations would be a useful addition. 

Response: We appreciate your comments and have include in discussion section

Comment 4: Conclusion is only one sentence. Maybe add details of what may make a good study based on your findings. 

Response: We appreciate your comment and agree with the lack of information. We have rewritten the conclusion section

Round 2

Reviewer 2 Report

Comments and Suggestions for Authors

Dear Authors,

Thank you for the opportunity to review your manuscript for this second round. Below, I provide my feedback, structured by major and minor points.

Major concern

The manuscript describes the use of the GRADE assessment before conducting the meta-analysis. This sequence raises significant concerns about the validity of the certainty of evidence assessment. GRADE is designed to evaluate the quality of evidence by considering study-level attributes (e.g., risk of bias, inconsistency, imprecision) alongside the synthesized effect sizes calculated in the meta-analysis. However, in this study, only the certainty assessment was used for GRADE because it seems to be performed before calculating effect sizes, which means the GRADE evaluation was based solely on study characteristics without incorporating the synthesized outcomes.

This approach could undermine the ability of the GRADE assessment to reflect the strength of the evidence accurately along with the certainty profile. I recommend either revisiting the GRADE assessment after the meta-analysis or providing a clear justification for this methodological choice within the manuscript. 

Minor Concerns

  1. In the abstract, it is unnecessary to use numbers before each section. Removing them would improve readability and align with standard formatting practices.

  2. There are typographical errors in the manuscript. For instance, RevMan's citation contains a typo that should be corrected. 

  3. The Aim is to evaluate the "effectiveness" of interventions. However, the included studies seem more aligned with assessing "efficacy," as they evaluate interventions in controlled settings. Effectiveness refers to how well an intervention works in real-world settings, whereas efficacy pertains to its performance in controlled conditions. While the choice of terms is ultimately yours, I recommend clarifying this distinction to ensure accuracy.

  4. The manuscript refers to PRIMS but does not provide sufficient detail about its use in the Methods section. The classification of interventions using PRIMS should be explicitly stated with an appropriate reference. Additionally, "PRIMS" should be spelled out in full upon its first mention in the text.

  5. There are several punctuation errors throughout the manuscript. Citations in the text should be placed before punctuation marks. Please carefully review and revise the manuscript for similar issues to ensure consistency.

Author Response

Please, find enclosed a revision of our manuscript, “Effects on Quality of Life and Self-efficacy of Instant Messaging Services in Self-Management Programs for Prostate Cancer: A Systematic Review and Meta-Analyses.”. We would like to thank the Editorial committee for giving us the opportunity to revise and improve our manuscript, and the Editors and reviewers for their thoughtful and constructive comments. We have considered all of the suggestions and have incorporated them into the revised manuscript. All changes to the manuscript have been highlighted or indicated by using tracked changes, and we believe our manuscript is stronger as a result of these modifications. An itemized point-by-point response to the reviewers’ comments is presented below.

Reviewer 1

Very Respected Authors,

Dear Authors,

Thank you for the opportunity to review your manuscript for this second round. Below, I provide my feedback, structured by major and minor points.

Major concern

Comment 1: The manuscript describes the use of the GRADE assessment before conducting the meta-analysis. This sequence raises significant concerns about the validity of the certainty of evidence assessment. GRADE is designed to evaluate the quality of evidence by considering study-level attributes (e.g., risk of bias, inconsistency, imprecision) alongside the synthesized effect sizes calculated in the meta-analysis. However, in this study, only the certainty assessment was used for GRADE because it seems to be performed before calculating effect sizes, which means the GRADE evaluation was based solely on study characteristics without incorporating the synthesized outcomes. This approach could undermine the ability of the GRADE assessment to reflect the strength of the evidence accurately along with the certainty profile. I recommend either revisiting the GRADE assessment after the meta-analysis or providing a clear justification for this methodological choice within the manuscript.

Response: We appreciate your comment and accept the error and have made several changes to our manuscript: First, we have modified the order of expression in both methodology and results to make it easier for readers to understand since data from the meta-analysis are used in the GRADE scale. As you have indicated, we have taken into account the effect size in our results and we have reflected it in the methodology and results section.

Minor Concerns

Comment 2: In the abstract, it is unnecessary to use numbers before each section. Removing them would improve readability and align with standard formatting practices.

Response: We appreciate your comment and agree that it helps to clarify and facilitate the reading of the manuscript. For this reason, we have eliminated the numbering

Comment 3: There are typographical errors in the manuscript. For instance, RevMan's citation contains a typo that should be corrected.

Response: Thank you for your comment and we have modified the error and other typographical errors in the text.

Comment 4: The Aim is to evaluate the "effectiveness" of interventions. However, the included studies seem more aligned with assessing "efficacy," as they evaluate interventions in controlled settings. Effectiveness refers to how well an intervention works in real-world settings, whereas efficacy pertains to its performance in controlled conditions. While the choice of terms is ultimately yours, I recommend clarifying this distinction to ensure accuracy.

Response: We appreciate your comment and agree that our review is clearer with the term “efficacy” and therefore we have modified it in the manuscript.

Comment 5: The manuscript refers to PRIMS but does not provide sufficient detail about its use in the Methods section. The classification of interventions using PRIMS should be explicitly stated with an appropriate reference. Additionally, "PRIMS" should be spelled out in full upon its first mention in the text.

Response: We appreciate your comments and have clarified the taxonomy in the methodology section.

Comment 6: There are several punctuation errors throughout the manuscript. Citations in the text should be placed before punctuation marks. Please carefully review and revise the manuscript for similar issues to ensure consistency.

Response: Thank you for your comment and we have modified all the references in the text.

Round 3

Reviewer 2 Report

Comments and Suggestions for Authors

Thank you for addressing the initial feedback and improving the clarity and robustness of the manuscript. The revised version shows advancements in methodological explanation and interpretation of results. However, by reading again the manuscript, a critical methodological issue remains unaddressed regarding the computation of the effect size (mean difference, MD) in the meta-analysis using RevMan.

The primary issue is related to the methodological appropriateness of using the mean difference (MD) across outcomes derived from varying measurement instruments. While the constructs (quality of life and self-efficacy) are consistent across studies, the tools used to assess these outcomes differ substantially in the primary included studies as per what is my understanding of table 2. These tools vary in their scales and scoring systems, which violates the assumptions underpinning the use of MD. Using MD in such cases does not allow for meaningful comparisons across studies, as the absolute values are not directly comparable.

To address this, the analysis should use the standardized mean difference (SMD) instead of MD. The SMD adjusts for differences in scale across measurement tools by standardizing effect sizes. This adjustment ensures that the results are comparable and may be synthesized appropriately, even when different instruments are used across studies. 

Author Response

Please, find enclosed a revision of our manuscript, “Effects on Quality of Life and Self-efficacy of Instant Messaging Services in Self-Management Programs for Prostate Cancer: A Systematic Review and Meta-Analyses.”. We would like to thank the Editorial committee for giving us the opportunity to revise and improve our manuscript, and the Editors and reviewers for their thoughtful and constructive comments. We have considered all of the suggestions and have incorporated them into the revised manuscript. All changes to the manuscript have been highlighted or indicated by using tracked changes, and we believe our manuscript is stronger as a result of these modifications. An itemized point-by-point response to the reviewers’ comments is presented below.

Reviewer 1

Comment 1: Thank you for addressing the initial feedback and improving the clarity and robustness of the manuscript. The revised version shows advancements in methodological explanation and interpretation of results. However, by reading again the manuscript, a critical methodological issue remains unaddressed regarding the computation of the effect size (mean difference, MD) in the meta-analysis using RevMan.

The primary issue is related to the methodological appropriateness of using the mean difference (MD) across outcomes derived from varying measurement instruments. While the constructs (quality of life and self-efficacy) are consistent across studies, the tools used to assess these outcomes differ substantially in the primary included studies as per what is my understanding of table 2. These tools vary in their scales and scoring systems, which violates the assumptions underpinning the use of MD. Using MD in such cases does not allow for meaningful comparisons across studies, as the absolute values are not directly comparable.

To address this, the analysis should use the standardized mean difference (SMD) instead of MD. The SMD adjusts for differences in scale across measurement tools by standardizing effect sizes. This adjustment ensures that the results are comparable and may be synthesized appropriately, even when different instruments are used across studies.

Response: We appreciate your comment and we fully agree with what you indicate, in fact the first version of the manuscript that was sent presented the meta-analysis using SMD instead of MD, however one of the reviewers demanded the change. Nevertheless, we resubmit our first version of the meta-analysis and leave it reflected in the methodology and results section.